# Investigating the Impact of IL6 on Insulin Secretion: Evidence from INS-1 Cells, Human Pancreatic Islets, and Serum Analysis

**DOI:** 10.3390/cells13080685

**Published:** 2024-04-15

**Authors:** Jalal Taneera, Anila Khalique, Abdul Khader Mohammed, Bashair M. Mussa, Nabil Sulaiman, Eman Abu-Gharbieh, Waseem El-Huneidi, Maha M. Saber-Ayad

**Affiliations:** 1College of Medicine, University of Sharjah, Sharjah P.O. Box 27272, United Arab Emirates; bmussa@sharjah.ac.ae (B.M.M.); nsulaiman@sharjah.ac.ae (N.S.); eabugharbieh@sharjah.ac.ae (E.A.-G.); welhuneidi@sharjah.ac.ae (W.E.-H.); msaber@sharjah.ac.ae (M.M.S.-A.); 2Research Institute of Medical and Health Sciences, University of Sharjah, Sharjah P.O. Box 27272, United Arab Emirates; aabid@sharjah.ac.ae (A.K.); amohammed@sharjah.ac.ae (A.K.M.); 3School of Pharmacy, The University of Jordan, Amman 11942, Jordan

**Keywords:** IL6, IL6R, INS-1 cell line, insulin secretion, RNA-seq, siRNA, human islets, type 2 diabetes

## Abstract

Interleukin-6 (IL6) is a pleiotropic cytokine implicated in metabolic disorders and inflammation, yet its precise influence on insulin secretion and glucose metabolism remains uncertain. This study examined *IL6* expression in pancreatic islets from individuals with/without diabetes, alongside a series of functional experiments, including siRNA silencing; IL6 treatment; and assessments of glucose uptake, cell viability, apoptosis, and expression of key β-cell genes, which were conducted in both INS-1 cells and human islets to elucidate the effect of IL6 on insulin secretion. Serum levels of IL6 from Emirati patients with type 2 diabetes (T2D) were measured, and the effect of antidiabetic drugs on IL6 levels was studied. The results revealed that *IL6* mRNA expression was higher in islets from diabetic and older donors compared to healthy or young donors. *IL6* expression correlated negatively with *PDX1*, *MAFB*, and *NEUROD1* and positively with *SOX4*, *HES1*, and *FOXA1*. Silencing IL6 in INS-1 cells reduced insulin secretion and glucose uptake independently of apoptosis or oxidative stress. Reduced expression of *IL6* was associated with the downregulation of *Ins*, *Pdx1*, *Neurod1*, and *Glut2* in INS-1 cells. In contrast, IL6 treatment enhanced insulin secretion in INS-1 cells and human islets and upregulated insulin expression. Serum IL6 levels were elevated in patients with T2D and associated with higher glucose, HbA1c, and triglycerides, regardless of glucose-lowering medications. This study provides a new understanding of the role of IL6 in β-cell function and the pathophysiology of T2D. Our data highlight differences in the response to IL6 between INS-1 cells and human islets, suggesting the presence of species-specific variations across different experimental models. Further research is warranted to unravel the precise mechanisms underlying the observed effects of IL-6 on insulin secretion.

## 1. Introduction

The underlying pathophysiology of diabetes mellitus (DM) remains not fully understood. However, it hinges on the failure of the insulin-producing pancreatic β-cells that play a pivotal role in maintaining glucose equilibrium [1]. Insulin secretion is regulated by various factors, including cytokines, growth factors, and hormones [2,3,4,5]. Among these, interleukin-6 (IL6) emerges as a cytokine secreted by diverse cell types ranging from T, B, monocytes, and macrophages to adipocytes [6]. The pleiotropic nature of IL6 becomes evident as it has dual roles, acting both as a pro-inflammatory agent and an anti-inflammatory mediator [7,8]. IL6 has been implicated in various physiological and pathological processes, including immune responses, inflammation, and glucose metabolism [9]. In addition, its involvement in the pathogenesis of chronic complications of the latter was also proposed and investigated [10].

IL6 exerts its effects through binding to its receptor, IL6R, and activating a complex signaling pathway that modulates the expression and activity of various transcription factors [11,12]. Previous studies have suggested that IL6 may play a role in insulin secretion and glucose metabolism, but the results are inconsistent and contradictory. Some studies have reported that IL6 stimulates insulin secretion in β-cells through different mechanisms, such as the PLC–IP3 pathway, the AMPK pathway, the Ca^2+^ influx, and GLP-1 expression [13,14,15,16]. Other studies have shown that IL6 protects β-cells from apoptosis induced by other cytokines, such as IL-1β and TNFα, possibly by inducing autophagy [17,18]. Moreover, IL6 has been implicated in regulating insulin resistance and obesity, as genetic deletion of IL6 in mice leads to obesity and glucose intolerance [19], and elevated serum IL6 levels are associated with insulin resistance in patients with T2D and obesity [4,20,21]. However, other studies have failed to find any effect of IL6 on insulin secretion in β-cells under diabetic-like conditions [22], or have suggested that IL6 may indirectly affect insulin secretion by increasing GLP-1 expression in α-cells [16]. Therefore, the role of IL6 in insulin secretion and glucose metabolism remains unclear and controversial. 

In this study, we aim to shed more light on the role of IL6 in insulin secretion and glucose metabolism by exploring the expression patterns of IL6 and IL6R in human pancreatic islets from donors with/without diabetes and their correlation with HbA1c, BMI, age, and gender. The effect of IL6 on insulin secretion and the expression of β-cell markers in rat INS-1 cells and human pancreatic islets was investigated using siRNA silencing and IL6 treatment models. Furthermore, we measured the serum levels of IL6 in Emirati subjects with/without T2D and their association with age and BMI. Finally, the influence of glucose-lowering therapy on the serum levels of IL6 was investigated.

## 2. Materials and Methods

### 2.1. Transcriptomic Analysis of Human Pancreatic Islets

Publicly accessible resources were used to retrieve RNA-seq expression data from human islets (n = 77 cadaver donors). The indispensable data were sourced from the NCBI Gene Expression Omnibus (GEO) database under the accession number GSE50398 [23]. Based on the HbA1c %, donors were stratified into two groups, comprising 61 nondiabetic/normoglycemic (age 77 ± 11.9, BMI 25.5 ± 2.2, HbA1c 5.3 ± 0.3) and 12 diabetic (age 61 ± 8.4, BMI 27.3 ± 3.8, HbA1c 6.7 ± 0.9) individuals.

### 2.2. Human Sampling and Data Collection

Two-hundred and forty-eight individuals from the United Arab Emirates National Diabetes Study (UAEDIAB) were randomly recruited for this study [24]. Subjects were stratified into 90 non-diabetic controls and 158 with T2D. All individuals with no history of cardiovascular diseases (CVD), dyslipidemia, or hyperglycemia were considered non-diabetic controls. In contrast, individuals with fasting glucose levels (126 mg/dL) or HbA1c levels (6.5%) were defined as diabetic based on the WHO criteria. Following the American Heart Association criteria, we used a 150 mg/dL cut-off and 200 mg/dL (triglycerides and total cholesterol) for dyslipidemia conditions. Individuals with BMI values of over 30.0 kg/m^2^ were defined as obese, those with 18.5 to 24.9 kg/m^2^ were considered average weight, and those with 25.0 to 29.0 kg/m^2^ were overweight. The ethical approval for using patient samples was obtained by the Ethics Review Board of the University of Sharjah and the Research Ethics Committee of the Ministry of Health and Prevention (MO-HAP/DXB/SUBC/No.14/2017). All participants provided informed consent before sample collection or access to medical records. Serum samples were stored at −80 °C for further testing. Analysis of FBG, HbA1c, and lipid profile (including Triglycerides, LDL-cholesterol, and HDL-cholesterol) was conducted using chemistry auto-analyzers at the clinical laboratory of the Rashid Center for Diabetes and Research in Ajman, UAE. Serum IL6 levels were measured using a commercially available ELISA assay (Elabscience, Wuhan, China). 

### 2.3. INS-1 Cell Culture Condition and Treatments

The rat insulinoma INS-1 (832/13) pancreatic β-cells (AddexBio, San Diego, CA, USA) were cultured and maintained in RPMI 1640 as previously described [25]. To assess the impact of IL6 on pancreatic INS-1 cells, we initially cultured the cells in 6-well plates overnight and then treated them with 200 ng/mL recombinant rat IL6-RRIL65 (Invitrogen, Waltham, MA, USA) for 48 h. 

### 2.4. Human Islets Culture Condition and Treatments

Human islets were procured from PRODO laboratories (Irvine, CA, USA). These islets were cultured in uncoated Petri dishes with PIMS media (Cat no. PIM-S001GMP). The impact of IL6 treatments on the islets was evaluated by exposing the islets to 200 ng/mL recombinant human IL6 (ab259381 Abcam, Cambridge, UK) for 48 h. Islets were then collected and subjected to mRNA and protein extraction. 

### 2.5. siRNA Transfection

INS-1 (832/13) cells were plated in a 24-well plate at a density of 200,000 cells/well in an antibiotic-free RPMI 1640 medium and transfected with siRNA sequence for IL6 (s127950; Thermo Fisher Scientific, Waltham, MA, USA), or scramble negative control siRNA, as was performed previously [25]. Insulin secretion, mRNA, and protein analysis were performed 48 h post-transfection.

### 2.6. Insulin Release Assay

Insulin secretion assay was performed as previously described [25]. Briefly, transfected or IL6-exposed INS-1 cells in 24-well plates were incubated in a secretion assay buffer (SAB) (114 NaCl; 4.7 KCl; 1.2 KH_2_PO_4_; 1.16 MgSO_4_; 20 HEPES; 2.5 CaCl_2_; 25.5 NaHCO_3_; 0.2% bovine serum albumin, pH 7.2–7.4) containing 2.8 mM glucose for 2 h. Following this, cells were stimulated for 1 h with SAB containing 2.8 mM or high glucose (16.7 mM). The amount of insulin released in the supernatants was measured using a rat insulin ELISA kit (Mercodia, catalog number Cat#:10-1145-01, Uppsala, Sweden). After exposing the human islets to IL-6 (10 islets per well, with 3 replicates), they were placed in SAB buffer with 1 mM glucose for one hour. The islets were then exposed to SAB buffer with either 2.8 mM or 16.7 mM glucose for an additional hour. A human ELISA kit measured insulin release in human islets (Mercodia, Sweden). The total cellular protein was extracted and diluted for insulin content measurement at 1:250. The insulin content was then assessed using an ELISA assay. Finally, results were normalized to the total protein amount in the cells.

### 2.7. Western Blot Analysis

The Western blot procedure was carried out as previously described [25]. In short, 40 μg protein lysate underwent separation using 12% sodium dodecyl sulfate-polyacrylamide gel electrophoresis (SDS-PAGE) and then was transferred onto a PVDF membrane. The membranes were then blocked for one hour using 5% skimmed milk. Then, the blots were exposed to primary antibodies against INS (Cat. #8138; Cell Signaling Technology, Danvers, MA, USA), INSRβ (Cat. #ab69508; Abcam), PDX1 (Cat. #5679, Cell Signaling Technology), GLUT2 (Cat. #A12307; ABclonal, Woburn, MA, USA), NEUROD1 (Cat. #A1147; ABclonal), MAFA (Cat. #ab264418), GCK (Cat. #ab37796), VAMP2 (Cat. #13508), and SNAP25 (Cat. #MA5-17609; Thermofisher, Waltham, MA, USA) overnight at 4 °C. The β-actin (Cat. #A5441; Sigma, St. Louis, MO, USA) was used as an endogenous control. Following the wash with TBST, the blots were then incubated with secondary antibodies, HRP-linked anti-mouse (#7076S, Cell Signaling, Danvers, MA, USA), and HRP-linked anti-rabbit (#7074; Cell Signaling) for 1 h at room temperature. Chemiluminescence was recorded using the Clarity ECL substrate kit and ChemiDoc Touch Gel Western Blot Imaging System (Bio-Rad, Hercules, CA, USA). The protein bands were quantified using the Bio-Rad Image Lab software (6.1).

### 2.8. Quantitative Real-Time Polymerase Chain Reaction (qPCR)

Total RNA was isolated using the RNA purification kit from NORGEN (Cat. #17200; NORGEN Biotek. Thorold, ON, Canada). Reverse transcription and cDNA were synthesized with the high-capacity reverse transcription kit (Applied Biosystems, Waltham, MA, USA). TaqMan gene expression assays for *Glut2* (Rn00563565_m1), *Ins1* (Rn02121433_g1), *Ins2* (Rn01774648_g1), *Pdx1* (Rn00755591_m1), *Insr* (Rn00690703_m1), *Gck* (Rn00561265_m1), and Rat *Hprt1* (Rn01527840_m1) were used to determine mRNA expression. At the same time, SYBR green chemistry was used to quantify Snap25, Vamp2, Mafa, and Neurod1 expression. The corresponding sequence for the SYBR primers utilized are presented in Table 1. The *Hprt1* gene was used as an endogenous control for normalizing the expression of target mRNA. All qPCR reactions were performed in triplicate in 96-well plates using the QuantStudio3 qPCR system (Applied Biosystems, USA). Fold change in mRNA expression was determined using the 2^−ΔΔCt^ method.

### 2.9. Cell Viability and Apoptosis Assay

Cell viability was assessed utilizing MTT assay reagents (Sigma-Aldrich, Burghausen, Germany). The cell apoptosis levels were examined using the Annexin V-FITC test, following the protocol outlined in a previous publication [26].

### 2.10. Glucose Uptake

The glucose uptake levels in transfected or IL6-treated cells were investigated using the 2-NBDG assay (Invitrogen, USA), following the methodology outlined in our prior studies [27]. Initially, cells were exposed to the 2-NBDG solution (100 µM/1 mL medium) and incubated for 1 h at 37 °C. Subsequently, the cells were trypsinized and incubated in 200 µL of cold PBS. Finally, flow cytometry analysis was conducted using excitation at 465 nm and emission at 540 nm (FACS AriaTM III). 

### 2.11. Reactive Oxygen Species (ROS) Measurements

The ROS production levels in transfected cells were quantified using the H_2_O_2_ assay (ROS-Glo H_2_O_2_ Assay-Promega, Madison, WI, USA), following the manufacturer’s instructions as described previously [27]. Initially, 20 µL of the H_2_O_2_ substrate solution was added to the transfected cells and incubated for four hours, followed by the addition of 100 µL of the ROS-Glo detection solution for 20 min. Subsequently, luminescence was measured using a plate reader, and the relative luminescence units (RLUs) were determined. 

### 2.12. Statistical Analysis

To measure the serum level of IL6, we used an independent sample *t*-test and ANOVA to compare the differences between the studied groups. In contrast, the Mann–Whitney U test was used to compare nonparametric variables. All data analysis was conducted using the Statistical Package for Social Sciences version 26.0 (SPSS Inc., Chicago, IL, USA). Statistical significance levels were set at *p* < 0.05. For RNA-seq, qPCR, or Western blot analysis, we performed Spearman’s correlation test to assess the association between gene expression and metabolic phenotypes. The non-parametric Mann–Whitney test assessed significance levels in differential gene expression data. Student’s *t*-test was used to analyze the differences in qPCR and Western blot. Statistical analyses were performed using Prism (GraphPad, V8). Differences were considered significant at * *p* < 0.05.

## 3. Results

### 3.1. RNA-Seq Expression Analysis of IL6 and IL6R in Human Pancreatic Islets

To analyze the mRNA expression patterns of IL6 and its receptor, IL6R, in human pancreatic islets, we utilized publicly available RNA-seq data comprising diabetic and non-diabetic groups. As depicted in Figure 1A, our findings reveal the presence of IL6 and IL6R transcripts in non-diabetic human pancreatic islets (n = 68). Notably, IL6 displayed a significantly higher expression level (*p* < 0.05) than IL6R. When we compared the expression of IL6 to that of other crucial β-cell marker genes, IL6 exhibited higher expression (*p* < 0.05) than KCNJ11 and MAFA while showing lower expression than PDX1, SNAP25, NEUROD1, and VAMP2 (Figure 1A). GCK, GLUT1, and INSR exhibited expression levels comparable to IL6 (Figure 1A). Additionally, we expanded our analysis to examine the expression of IL6 and IL6R in various metabolic tissues, including fat, islets, liver, and muscle, utilizing the Islet Gene View (IGV) web tool (https://mae.crc.med.lu.se/IsletGeneView/ (12 January 2024) [28]. As illustrated in Figure 1B, IL6 expression was most pronounced in human islets and fat tissue compared to the liver and muscle. In contrast, IL6R exhibited the highest expression in muscle and the lowest in fat tissues (Figure 1C). Moreover, the expression of IL6 and IL6R showed no significant correlation in human islets (Figure 1D).

Conducting a differential mRNA expression analysis of IL6 and IL6R in human islets, we observed that IL6 was significantly upregulated (*p* < 0.05) in diabetic islets compared to non-diabetic ones (Figure 1E). However, mRNA expression of IL6R remained unaffected by the diabetic status (not shown). Further examination of IL6 expression based on donor age in diabetic/non-diabetic donors aged above 50 years of age exhibited higher expression (*p* < 0.05) than in donors (diabetic/non-diabetic) below 50 years (Figure 1F). No significant differences in IL6 expression were observed based on gender or body mass index (BMI) (Figure 1G,H). Together, our analysis elucidated a varied mRNA expression of IL6 and IL6R in human pancreatic islets and other metabolic tissues, with IL6 notably upregulated in diabetic islets and older donors. This suggests potential implications for age-related pancreatic dysfunction and diabetes.

### 3.2. Correlation of IL6 Expression and Genes Involved in the Function of Pancreatic β-Cells

Next, using Spearman’s correlation method, we systematically investigated the relationship between IL6 and genes critical for β-cell function. Our initial focus was on examining the expression correlation between the IL6 and 41 transcription factors integral to pancreatic β-cell functionality. This analysis revealed a diverse range of associations: 19 transcription factors were negatively correlated with IL6, 9 demonstrated a positive correlation, and 13 showed no significant correlation at the *p* < 0.05 level (Figure 2A). Notably, PDX1, MAFB, and NEUROD1 emerged as the top three transcription factors with robust negative correlations (r − 0.7; *p* < 0.000), indicating a strong inverse relationship with IL6 levels. In contrast, SOX4, HES1, and FOXA1 were identified as the leading transcription factors positively correlated with IL6 (r + 0.4; *p* < 0.0001), suggesting a different regulatory interaction. Further expanding our analysis, we explored the co-expression patterns of IL6 with 33 genes associated with the exocytosis machinery. This examination yielded varied correlations: 13 genes were negatively correlated, 3 genes showed a positive correlation, and 17 genes had no significant correlation (Figure 2B). SYT17, SYT13, and SYT14 stood out as the top three genes displaying robust negative correlations (r − 0.5; *p* < 0.0001). Conversely, SNAP23, SYT12, and SYT16 were the top genes exhibiting positive correlations (r > −0.3; *p* < 0.0001). Moreover, we delved into the expression correlation between IL6 and 13 genes involved in intracellular calcium transport. This part of our study revealed that eight genes were negatively correlated with IL6. In contrast, the remaining genes demonstrated no significant correlation (Figure 2C). The genes CACNB2, CACNA2D2, and CACNA2D1 were identified as having the strongest negative correlations (r > 0.4; *p* < 0.0001). 

It is well accepted that various factors, including glucotoxicity, lipotoxicity, steroid hormones, and medications, are presumed to influence gene expression [29,30,31]. Therefore, we evaluated the effects of these factors on the expression of *IL6* mRNA in INS-1 cells. To this end, cells were subjected to short-term (24 h) exposure to high glucose concentrations (25 mM), palmitic acid (PA) (200 μM), dexamethasone (100 nM), insulin (10 μM), metformin (10 μM), and Rosiglitazone (25 μM). As depicted in Figure 2D, exposure to high concentrations of glucose, dexamethasone, or PA for 24 h significantly elevated IL6 expression (*p* < 0.05). No effects were observed on IL6 expression following treatment with the antidiabetic medications. In summary, our analysis underscores the complex and varied nature of IL6’s interactions with genes that play pivotal roles in pancreatic β-cell function, encompassing transcription regulation, exocytosis, and calcium transport. Increased expression of IL6 in diabetic islets could also be attributed to metabolic stressors.

### 3.3. Effect of IL6 Silencing on Insulin Secretion and Glucose Uptake in INS-1 Cells

To study the precise role of IL6 in the pancreatic beta cells, we eliminated the expression of IL6 in INS-1 cells to elucidate its role in β-functions, insulin secretion, glucose uptake, and gene expression regulation. Examination of mRNA expression of IL6 48-h post-transfection showed a marked reduction of approximately 75% (*p* < 0.05) in comparison to the siRNA negative control (Figure 3A). Importantly, this pronounced decrease in *IL6* expression did not impact cell viability, as tested by the MTT assay results (Figure 3B). Moreover, no discernible alterations were detected in the apoptosis rate, measured through the annexin-V assay (Figure 3C), nor in the levels of ROS, as tested using the ROS-Glo H_2_O_2_ assay (Figure 3D). Nevertheless, a noteworthy finding emerged, with a significant reduction of approximately 25% (*p* < 0.05) in the rate of glucose uptake observed in *IL6*-silenced cells compared to their control counterparts (Figure 3E). Furthermore, our investigations revealed a substantial decrease (approximately 35%; *p* < 0.05) in insulin secretion from IL6-silenced cells following stimulation with 16.7 mM glucose (Figure 3F). Finally, the insulin content in IL6-silenced cells exhibited a significant reduction (*p* < 0.05) compared to the negative control cells (Figure 3G). These findings emphasize the important role of *IL6* in regulating insulin release and glucose uptake.

### 3.4. Impcat of IL6 Silencing on the Expression of Essential Genes Involved in β-Cell Function

Measuring gene expression in pancreatic β-cells provides insights into glucose homeostasis. For example, Ins, Pdx1, Neurd1, Mafa, Glut2, Gck, Insrβ, Snap25, and Vamp2 are key genes involved in regulating insulin production, secretion, and cell function. Dysregulation of these genes can lead to diabetes, emphasizing their importance in β-cell biology. Therefore, the impact of IL6 silencing on the expression of key genes for β-cell function was investigated at the transcriptional and translational levels in INS-1 cells. As illustrated in Figure 4A, mRNA expression of *Ins2*, *Pdx1*, *NeuroD1*, *Glut2*, and *Snap25* was significantly decreased (*p* < 0.05) in IL6-silenced cells compared to the control cells. Expression of *Ins1*, *Mafa*, *Vamp2* and *Gck* was not affected (Figure 4A). At protein expression, Pro/INS (~25%; *p* < 0.05), PDX1 (~40%; *p* < 0.05), NEUROD1 (~50%; *p* < 0.05), and GLUT2 (~50%; *p* < 0.05) were downregulated in IL6-silenced cells compared to in control cells (Figure 4B). In contrast, no significant changes were evident in the expression of MAFA, INSRβ, SNAP25, or VAMP2 upon IL6 silencing. 

### 3.5. Effect of IL6 Treatment on the Function of Pancreatic β-Cells

Having established the impact of IL6 silencing on β-cell biology, we further investigated the effect of IL6 treatment on INS-1 cells using recombinant protein to assess pancreatic β-cell function. As depicted in Figure 5, rat recombinant protein–IL6 treatment for 24 h showed no influence on cell viability (Figure 5A), ROS levels (Figure 5B), apoptosis rate (Figure 5C), or glucose uptake (Figure 5D) compared to control cells. Notably, a significant increase (*p* < 0.05) in insulin secretion was noticed upon IL6 treatment following stimulation with 16.7 mM glucose (Figure 5E). This was accompanied by a significant increase (*p* < 0.05) in insulin content compared to the control cells (Figure 5F). 

Analysis of the impact of IL6 treatment in INS-1 cells on the key genes essential for β-cell function revealed a significant increase in the mRNA expression of *Ins1* and *Mafa* (*p* < 0.05) compared to control cells (Figure 6A). In contrast, *Ins2*, *Pdx1*, *NeuroD1*, *Glut2,* and *Gck* genes were not affected by IL6-treatment. At protein expression, only Pro/INS were significantly upregulated (~75%; *p* < 0.05) (Figure 6B). Although the expressions of PDX1 and MAFA showed a trend of upregulation, they did not reach a significant degree (Figure 6B). No changes were observed in the expression of NEUROD1, GLUT2, GCK, or INSRβ upon IL6 treatment.

### 3.6. Effect of IL6 Treatment in Human Islets and Measurements of Serum Levels in Patients with T2D

Having verified the enhancement of insulin secretion in INS-1 cells following IL6 treatment, we sought to validate this observation in human pancreatic islets after exposure to IL6 recombinant protein for 24 h. As shown in Figure 7A, glucose-stimulated insulin secretion in IL6-treated islets significantly increased after stimulation with 16.7 mM glucose (*p* < 0.05). No effect was observed at lower doses of glucose (2.8 mM). 

Next, we investigated the serum levels of IL6 in T2D subjects (n = 137) compared to non-diabetic subjects (n = 92). Anthropometric data of T2D patients and controls are given in Table 2. The mean age in T2D subjects was 54 years; BMI greater than 31.8 kg/m^2^; fasting glucose levels 9.7 mmol/L; HbA1c 7.7%; total cholesterol 4.4 mmol/L; HDL-Cholesterol 1.2 mmol/L; LDL-Cholesterol 2.6 (mmol/L); and Triglycerides 1.5 mmol/L. Anti-diabetic medication showed that 30% of all T2D subjects were on oral medication, 8% were on insulin therapy, 21% were on combined medications (oral and insulin), and 48% were on dietary treatment. The type, duration, and dose of oral anti-diabetic medications were not reported. Measurements of plasma levels of IL6 as an indirect and partial reflection of its expression within pancreatic β-cells were found to be significantly higher (adj *p* < 0.01) in T2D subjects (24.4 + 16.9 pg/mL) than healthy controls (34.1 + 21.1 pg/mL) (Figure 7B). No significant differences in IL6 levels were observed in T2D subjects on oral anti-diabetic therapy, combined therapy, or dietary treatment (Figure 7C). Next, we correlated IL6 levels with clinical variables in T2D subjects. Correlation tests were performed using different models to assess whether these clinical variables had independent associations with IL6 levels. In model 1 (unadjusted analysis), IL6 levels were associated (*p* < 0.05) positively with age, WC, fasting glucose, HbA1c, and triglyceride levels (Table 3). Following adjustment for age and gender in the analysis of model 2, the associations between IL6 levels and fasting glucose, HbA1c, and triglycerides remained positive, indicating that increased levels of IL6 were associated with higher blood sugar and lipids profiles. These findings suggest that IL6 may play a role in the pathophysiology of T2D, possibly through its associations with metabolic parameters.

Data are presented as mean ± standard deviation for normal continuous variables; # denotes log or Sqrt transformation of data prior to analysis. The superscript θ denotes continuous variables with non-Gaussian distribution, which are presented as medians (1st–3rd quartile). The independent sample *t*-test and a Mann–Whitney U test were used to test differences between the control and T2DM groups.

## 4. Discussion

The present study provides insight into the potential role of IL6 in insulin secretion and pancreatic β-cell function. At first, we explored the expression of IL6 and IL6R in human pancreatic islets and other metabolic tissues, as well as their correlation with diabetes status, using publicly available datasets. The results revealed higher expression of IL6 in human islets and fat cells compared to the liver and skeletal muscles. In contrast, IL6R was highly expressed in the muscle and liver compared to islets and fat cells. This finding highlights that IL6 and IL6R are expressed differentially across different tissues and organs, as demonstrated by previous research [32]. No correlation between IL6 and IL6R in human islets was observed, suggesting that IL-6 does not have an autocrine function. However, it is likely to be secreted by the exhausted islet cells and adipocytes to act on remote tissues, e.g., the liver and skeletal muscles.

The observed expression patterns of IL6 and IL6R in different metabolic tissues likely reflect a distinct function and responses to various environmental and physiological cues, such as immune responses, inflammation, and metabolic processes. For example, IL-6 may be expressed in islets as part of the immune response or as a signaling molecule regulating insulin secretion or glucose metabolism. Fat cells secrete various signaling molecules, including cytokines like IL-6, as a part of their involvement in inflammation, metabolism, and energy homeostasis. The high expression of IL-6R in muscle may be related to its role in muscle function and repair. In contrast, it may be associated with regulating the response to inflammation or injury in the liver and muscles.

Our finding that IL6 expression is upregulated in pancreatic islets obtained from diabetic individuals (Figure 1E) is consistent with previous studies [33]. The finding reflects that inflammation is involved in the pathogenesis of T2D. This was supported by the findings that the islet histology of patients with T2D showed characteristic features of inflammation through the presence of cytokines, immune cell infiltration, and fibrosis [34]. This role could be attributed to the pro-inflammatory effect of IL6, which acts as an induction mechanism of islet inflammation associated with diabetes. The finding may also suggest that IL-6 plays a role in the pathophysiology of T2D, potentially contributing to hyperinsulinemia and perpetuation of the early stages of beta cell hyperfunction in the context of insulin resistance, as well as to β-cell exhaustion and overt diabetes [35,36].

Interestingly, recent reports have revealed an increased expression of IL6 in response to insulin administration in 3T3-L1 adipocytes [37]. Although these findings have been reported in adipose tissue, it may explain that increased expression of IL6 mRNA is a sequence of hyperinsulinemia associated with insulin resistance. To shed more light on this issue, we investigated the impact of exposure to glucose, lipids, steroid hormones, and antidiabetic medications on IL6 expression in INS-1 cells. It was evident that short-term exposure to high glucose levels, PA, or Dexamethasone for 24 h increased IL6 expression, denoting the role of IL6 as a mediator of the effect of those factors on β-cells. In contrast, antidiabetic medications (insulin, metformin, and Rosiglitazone) showed no impact on IL6 expression. These data indicate that the upregulation of IL6 in diabetic islets may be attributed to glucotoxicity, lipid toxicity, or steroid hormones rather than a direct cause. Our findings also revealed that IL6 expression was not changed under short-term insulin exposure; assuming that IL6 elevation may start during the early phases of insulin resistance (and hyperinsulinemia), longer exposure of insulin is needed to elevate IL6 expression, and treatment with different medications in vitro has a differential impact compared to in vivo administration.

The present study has assessed the IL6 serum levels based on various demographic, anthropometric, and clinical characteristics. In agreement with the present findings, several studies have shown that IL6 gene expression and serum levels increase with age [38,39]. Nevertheless, one report indicated that the IL6 level may rise with aging, but not beyond the normal range [40]. Although our current study used a younger age cutoff level (50 years), it revealed a significant elevation of IL6 in aged patients with T2D, denoting the ongoing inflammatory status in such patients, with a potential IL6 effect on exacerbating insulin resistance in those patients.

Our analysis failed to document differences in IL6 expression in pancreatic islets or in serum levels based on gender. However, it has been shown that women produce more IL-6 than men [41]. Moreover, we could not observe differential expression of IL6 in pancreatic islets between obese and lean donors, and we did not find any correlation of serum IL-6 with body mass index (BMI). Previous studies have shown positive correlations between serum IL6 and BMI in patients with morbid obesity [42].

Correlation analysis in human islet data has unveiled interesting insights into the relationship between IL6 expression and critical components of pancreatic β-cell function. Notably, IL6 exhibits an inverse correlation with 46% (19 out of 41) of a significant portion of key β-cell transcription factors, 40% (13 out of 33) of genes linked to the exocytosis machinery, and 60% (8 out of 13) of genes involved in intracellular calcium transport. This expression pattern suggests that heightened IL6 expression is linked to diminished expression of crucial transcription factors like PDX1; exocytosis-related genes such as SYT17; and genes associated with intracellular calcium transport, exemplified by CACNB2. These findings resemble a hallmark characteristic of dysfunctional pancreatic β-cells, characteristic of T2D. The pathogenesis of T2D is phasic, with a complex interplay of insulin resistance (characterized by hyperinsulinemia) and beta-cell dysfunction (leading ultimately to relative insulin deficiency). The correlation analysis in human islet data is in agreement with a previously published report showing that exposure of mouse islets to IL6 decreases mRNA expression of Ins2, Pdx1, Nkx6-1, Foxo1, and Slc2a2 [43]

Functional studies using rat INS-1 (832/13) cells provided further insights into the role of IL6 in β-cell function. Silencing of IL6 resulted in impaired insulin secretion, content, and glucose uptake without affecting apoptosis levels or ROS production, indicating the importance of physiological IL6 levels for optimal β-cell function. The impairment of insulin secretion was associated with the downregulation of key β-cell genes, including *Insulin*, *Pdx1*, *NeuroD1*, and *Glut2*, which further supported the crucial role of IL6 in maintaining normal β-cell function. *Pdx1* and *NeuroD1* are key transcription factors for β cell maturation and responsiveness to glucose through regulating the insulin gene expression. *Glut2* gene is the main glucose transport in rodent β-cells and is a vital component of the glucose uptake machinery for insulin release. In addition, we demonstrate that INS-1 cells and human islets treated with IL6 enhance insulin secretion. Mechanistically, IL6 treatment was associated with upregulation of insulin gene expression [15]. The opposite findings observed in this study between INS-1 cells and human islets can predominantly be attributed to species variation. Such variations suggest that some degree of species-specific traits may exist, or that there are disparities between freshly isolated normal cells and immortalized cell lines. Hence, selecting the appropriate experimental model is crucial when aiming to accurately characterize the functional aspects of genes/receptors and their impact on cell biology.

The findings of this study are consistent with previous research suggesting a multifaceted impact of IL6 on various aspects of β-cell function and glucose metabolism. Positive effects of IL6 on insulin secretion, β-cell signaling, metabolism, antioxidants, and protection from apoptosis induced by cytokines have been reported in previous studies [13,16,17,44]. However, it is important to acknowledge the conflicting results reported in the literature regarding the role of IL6 in insulin secretion [45,46,47]. Some studies have proposed indirect mechanisms through which IL6 enhances insulin secretion, and some studies have found no evidence supporting a direct role of IL6 in regulating β-cell function. These discrepancies may stem from variations in experimental conditions, research models, cell types, and concentrations of IL6 used in different studies.

The clinical data results demonstrated significantly higher serum levels of IL6 in Emirati T2D patients compared to controls. Moreover, IL6 serum levels were positively associated with FBG, HbA1c %, and triglycerides, suggesting a potential association between IL6 and the development of T2D. Importantly, no impact of glucose-lowering drugs on IL6 levels in diabetic patients was observed, indicating that IL6 dysregulation may occur independently of these treatments. These findings, to a high extent, are in line with our in vitro data on the impact of antidiabetic medications on IL6 expression in INS-1 cells. Notably, it has been reported that chronic elevation of IL6 levels negatively affects hepatic insulin sensitivity in rodents.

In contrast, acute IL6 infusion had no impact on nor did it positively affect human insulin sensitivity [48]. Additionally, IL6 inhibits insulin-induced glycogen synthesis and insulin receptor signal transduction in mouse hepatocytes and HepG2 cells by inhibiting insulin activation of Akt [49]. Another study showed that acute exposure to IL6 induces SOCS-3 (potential inhibitor of insulin signaling factor) mRNA and protein expression that directly inhibits insulin receptor autophosphorylation [36]. The IL6 antagonists are indicated in the treatment of rheumatological disorders and in patients with cytokine release syndrome, occurring as an adverse effect of chimeric antigen receptor T cell therapy [50]. Both tocilizumab and sarilumab are recombinant humanized anti-IL6 receptor monoclonal antibodies. In the context of diabetes mellitus, as mentioned earlier, IL-6 has been implicated in the development of insulin resistance, a hallmark of type 2 diabetes. Furthermore, the intricate interplay between chronic low-grade inflammation and insulin resistance, as well as obesity, underscores the significance of IL-6 in the pathophysiology and disease progression. Consequently, targeting IL-6 or its downstream signaling cascades emerges as a promising therapeutic strategy for effectively managing diabetes mellitus, with particular emphasis on T2D.

## 5. Conclusions

The present study highlights the significant association between IL6 and T2D, providing evidence for elevated serum levels of IL6 in T2D patients. The study further elucidates the role of IL6 in insulin secretion, apoptosis, and inflammation in INS-1 cells, emphasizing the importance of physiological IL6 levels for maintaining normal β-cell function.

Our data underscore variances in the reaction to IL6 between INS-1 cells and human islets, indicating the existence of discrepancies among various experimental models. These findings contribute to our understanding of the complex interplay between IL6 and T2D pathogenesis, paving the way for potential therapeutic interventions targeting IL6 in the future management of T2D.

## Figures and Tables

**Figure 1 cells-13-00685-f001:**
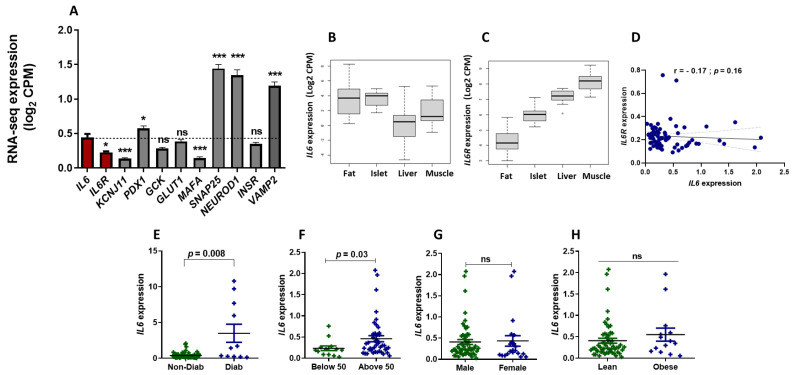
Analysis of IL6 and IL6R expression pattern in human pancreatic islets and metabolic tissues. (**A**) RNA-seq expression of IL6 and IL6R compared to other key pancreatic β-cell genes (KCNJ11, PDX1, GCK, GLUT1, MAFA, SNAP25, NEUROD1, INSR, and VAMP2) in non-diabetic human pancreatic islets (n = 68). (**B**,**C**) RNA-seq expression (acquired from IGV portal) of IL6 and IL6R in human fat tissue (n = 12), pancreatic islet (n = 12), liver (n = 12), and skeletal muscle tissue expression (n = 12) obtained from the same non-diabetic donors. (**D**) Co-expression correlation of IL6 with IL6R in human islets (n = 68) obtained from non-diabetic donors. (**E**) Expression of IL6 in human islets obtained from hyperglycemic, including diabetic, donors (n = 11) versus normoglycemic/non-diabetic donors (n = 11). (**F**) Expression of IL6 in donors ≥ 50 years old (n = 43) versus donors ≤ 40 years old (n = 13). (**G**) Expression of IL6 in males (n = 29) versus females (n = 21). (**H**) Expression of IL6 in lean donors (BMI ≤ 24; n = 50) versus obese donors (BMI ≥ 29; n = 14). The R and *p* values are displayed in each figure. CPM; counts per million. R; correlation coefficient. P; *p*-value. * *p* < 0.05, *** *p* < 0.001. ns, not significant. Bars above histograms represent the mean ± SD of the values.

**Figure 2 cells-13-00685-f002:**
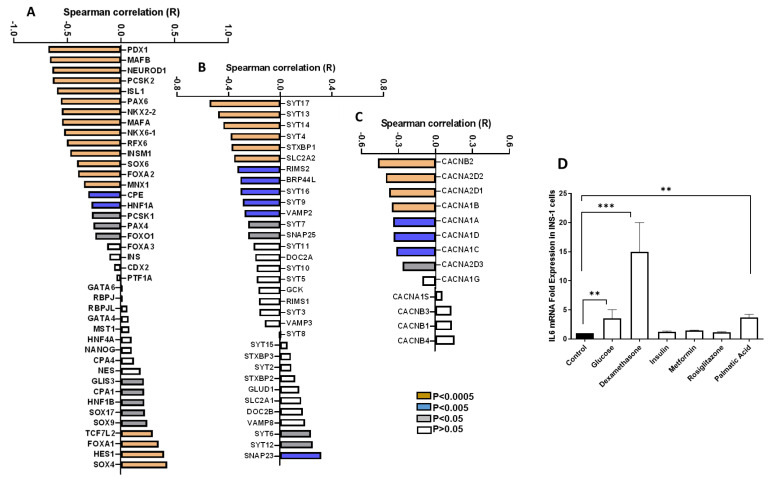
Correlation analysis of IL6 expression with key pancreatic β-cell genes in human islets. Spearman co-expression correlation of IL6 with key transcription factors in β-cells (**A**); genes involved in exocytosis machinery (**B**); and intracellular calcium transport genes (**C**) using RNA-seq expression data from human pancreatic islets (n = 88). (**D**) mRNA expression analysis of IL6 in INS-1 cell treated with high glucose (25 mM), palmitic acid (200 µM), dexamethasone (100 nM), metformin (10 µM), insulin (10 µg), or Rosiglitazone (25 µM) for 24 h. ** *p* < 0.01, *** *p* < 0.001. ns; not significant.

**Figure 3 cells-13-00685-f003:**
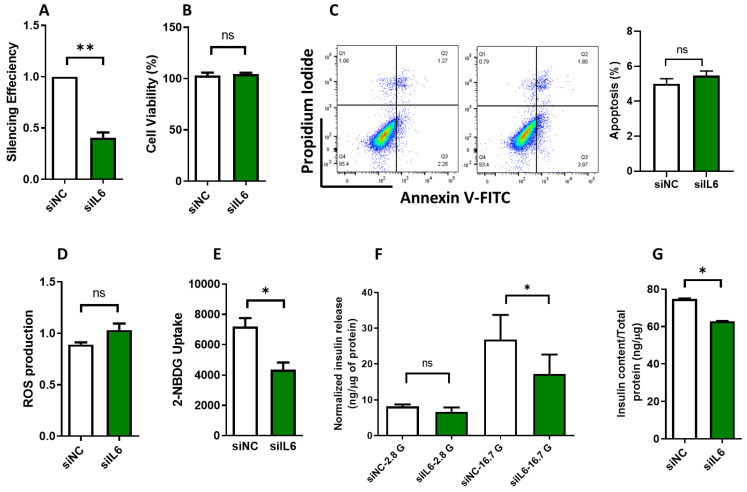
Silencing of IL6 expression causes β-cell dysfunction in INS-1 cells. (**A**) Analysis of the mRNA expression of IL6 in INS-1 cells after 48 h of siRNA silencing. (**B**) Cell viability assay determined by MTT test in IL6 or control siRNA-transfected cells. (**C**) Apoptosis rate measurements by flow cytometry using Annexin-V assay were analyzed in IL6 or control siRNA-transfected cells. The right panels represent the apoptosis quantification. (**D**) Measurements of ROS levels using fluorescence intensity in IL6 or control siRNA-transfected cells. (**E**) The efficiency of glucose uptake in IL6-silenced cells compared to control cells. (**F**) Insulin release measurements (normalized to protein content) in IL6-silenced cells compared to control cells after stimulation with 2.8 or 16.7 mM glucose for 1 h. (**G**). Insulin content measurements in IL6-silenced cells compared to control cells. Data are generated from three different experiments. * *p* < 0.05, ** *p* < 0.01, ns; not significant. Bars represent mean ± S.D. of the values.

**Figure 4 cells-13-00685-f004:**
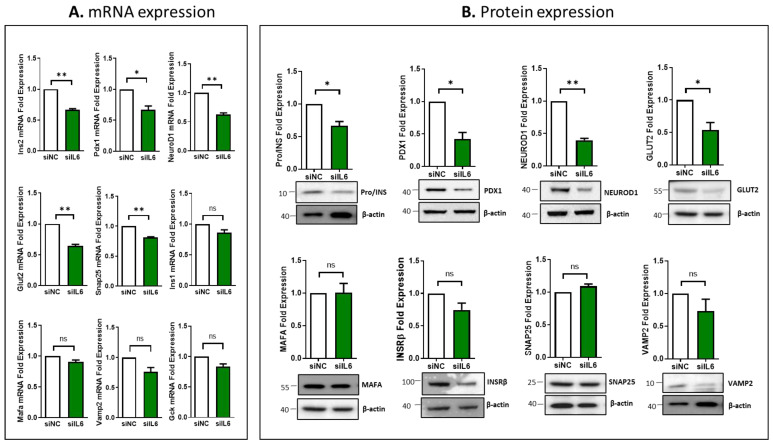
Silencing of IL6 modulates the expression of key β-cell function genes in INS-1 cells. (**A**) mRNA expression analysis of *Ins2*, *Pdx1*, *NeuroD1*, *Glut2*, *Snap25*, *Ins1*, *Mafa*, *Vamp2*, and *Gck* in IL6-silenced cells compared to control cells. (**B**) Western blot analysis expression analysis of Pro/insulin (INS), PDX1, NEUROD1, GLUT2, MAFA, INSRβ, SNAP25, and VAMP2 in IL6-silenced cells compared to control cells (lower panels). The fold changes of expression bands are shown in the upper panel. Data are generated from three different experiments. * *p* < 0.05, ** *p* < 0.01, ns; not significant. Bars represent mean ± S.D. of the values.

**Figure 5 cells-13-00685-f005:**
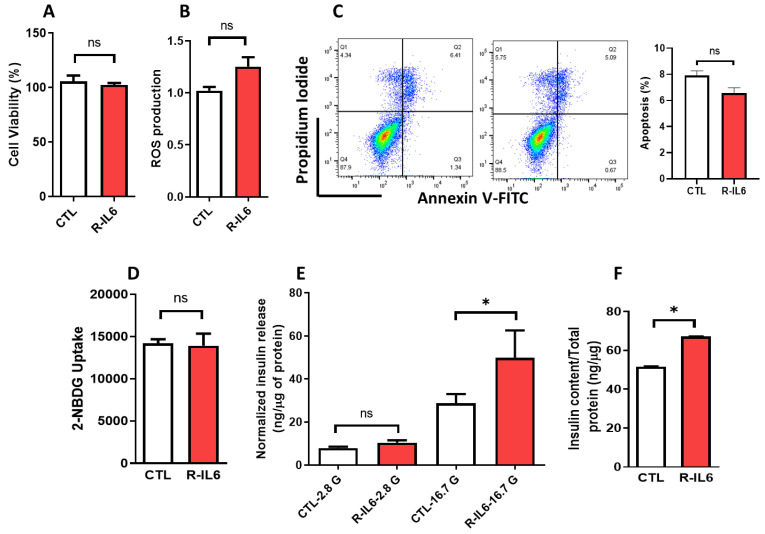
IL6-treatment enhanced insulin secretion in INS-1 cells. Rat recombinant protein-IL6 was used to treat INS-1 cells for 24 h, and then the cells were subjected to different tests compared to the control cells. (**A**) Cell viability assay. (**B**) ROS measurements. (**C**) Apoptosis rate measurements (the right panels represent the apoptosis quantification). (**D**) Glucose uptake efficiency. (**E**) Insulin release measurements (normalized to protein content) after stimulation with 2.8 or 16.7 mM glucose for 1 h. (**F**) Insulin content measurements. Data are generated from three different experiments. * *p* < 0.05, ns; not significant. Bars represent mean ± S.D. of the values.

**Figure 6 cells-13-00685-f006:**
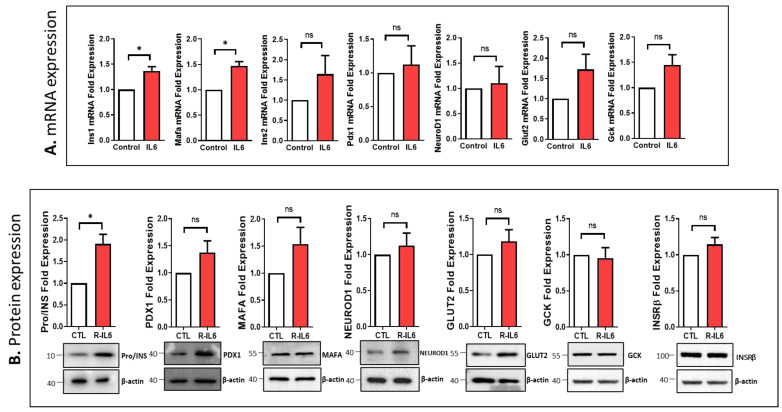
IL6 treatment modulated the expression of key β-cell function genes in INS-1 cells. (**A**) mRNA expression analysis of Ins1, Mafa, Ins2, Pdx1, NeuroD1, Glut2, and Gck in IL6-treated INS-1 cells compared to control cells. (**B**) Western blot analysis expression analysis of Pro/insulin (INS), PDX1, MAFA, NEUROD1, GLUT2, GCK, and INSRβ in IL6-treated cells compared to control cells (lower panels). The fold changes of expression bands are shown in the upper panel. Data are generated from three different experiments. * *p* < 0.05, ns; not significant. Bars represent mean ± S.D. of the values.

**Figure 7 cells-13-00685-f007:**
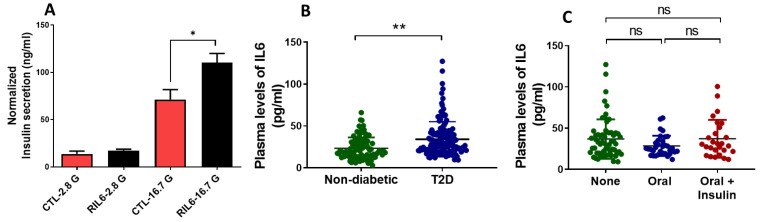
Impact of IL6 treatment in human islets and measurements of serum levels in diabetic subjects. (**A**) Insulin secretion measurements (normalized to protein content) in IL6-treated human islets compared to control islets stimulated with 2.8 or 16.7 mM glucose. Data were obtained from two non-diabetic donors (Prodo Laboratories INC, CA, USA). (**B**) Comparison of serum levels of IL6 in T2D subjects (n = 137) compared to healthy controls (n = 92). (**C**) Comparison of serum levels in subjects with no glucose-lowering medications (none) (n = 62), with oral medication (n = 34), or with combined oral and insulin (n = 27). Data are presented as median IL6 levels with interquartile ranges. * *p* < 0.05, ** *p* < 0.01. ns; not significant.

**Table 1 cells-13-00685-t001:** Primer sequence used for SYBR green chemistry.

Gene	Forward Sequence (5′-3′)	Reverse Sequence (5′-3′)
*Snap25*	GGCGTTTGCTGAATGACAAC	CAGAGCCTGACACCCTAAGA
*Vamp2*	TGGTGGACATCATGAGGGTG	GCTTGGCTGCACTTGTTTCA
*Mafa*	GAGGAGGAGCGCAAGATCGG	AGCAAAAGTTTCGTGCTGTCAA
*Neurod1*	CCC TAACTGATTGCACCAGC	TGCAGGGTAGTGCATGGTAA
*Hprt1*	TTGTGTCATCAGCGAAAGTGG	CACAGGACTAGAACGTCTGCT

**Table 2 cells-13-00685-t002:** Anthropometric and metabolic characteristics of the recruited subjects.

Variables (N)	Controls (99)	T2DM (134)	*p*-Value
Gender (Male/Female)	35/64	66/68	-
Smokers, n (%)	6 (6.9)	17 (13.6)	0.12
Age (Years)	36.4 ± 9.7	51.7 ± 13.7	<0.0001
BMI (kg/m^2^)	27.6 ± 4.8	31.3 ± 5.6	<0.0001
Waist circumference	88.7 ± 13.4	104.0 ± 15.2	<0.0001
SBP (mmHg)	118.0 ± 15.5	131.2 ± 21.2	<0.0001
DBP (mmHg)	77.5 ± 10.1	82.8 ± 10.4	<0.0001
Fasting Glucose (mmol/L)	5.2 ± 0.6	9.7 ± 4.1	<0.0001
HbA1c (%)	5.3 ± 0.4	7.7 ± 2.1	<0.0001
Total Cholesterol (mmol/L)	4.9 ± 0.9	4.6 ± 1.1	0.03
HDL-Cholesterol (mmol/L)	3.20 ± 0.83	2.85 ± 1.0	0.005
LDL-Cholesterol (mmol/L)	1.33 ± 0.34	1.16 ± 0.39	0.001
Triglycerides (mmol/L)	1.19 ± 0.71	1.77 ± 0.92	<0.0001
IL-6	24.4 ± 16.9	34.1 ± 21.1	<0.0001

**Table 3 cells-13-00685-t003:** Partial correlations between log serum IL-6 levels and various variables in T2DM subjects.

	Model 1		Model 2	
Characteristics	r	*p*-Value	r	*p*-Value
Age	0.26	<0.0001	-	-
BMI (kg/m^2^)	0.10	0.20	0.03	0.72
Waist circumference	0.25	<0.0001	0.07	0.35
SBP (mmHg)	0.08	0.23	0.003	0.97
DBP (mmHg)	0.008	0.90	−0.06	0.44
Fasting Glucose (mmol/L) #	0.22	0.001	0.18	0.02
HbA1c (%) #	0.30	0.002	0.25	0.001
Total Cholesterol (mmol/L)	−0.009	0.89	0.08	0.29
Triglycerides (mmol/L) #	0.36	<0.0001	0.35	<0.0001

Model 1: unadjusted. Model 2: adjusted for age and gender. # denotes log or Sqrt transformation of data prior to analysis.

## Data Availability

Data will be made available upon request.

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
