# Peer review of "Investigating the Impact of IL6 on Insulin Secretion: Evidence from INS-1 Cells, Human Pancreatic Islets, and Serum Analysis"

_cells, 2024, doi:10.3390/cells13080685_

Round 1

Reviewer 1 Report

Comments and Suggestions for Authors

Most of the work reported in the manuscript is already well known in rodent islets/cell lines and human islets and hence, its low in terms of novelty. Further, based on authors' secondary analyses Il6 shows negative correlation with Pdx1, neurod1 etc. But upon silencing Il6, surprisingly these genes are down regulated. Insulin secretion by the human islets is not normalized to insulin content.

Comments on the Quality of English Language

Article should be checked for grammar.

Author Response

First, we would like to thank the reviewer for the valuable comments and feedback.

REVIEWERS' COMMENTS:

Reviewer #1: Most of the work reported in the manuscript is already well known in rodent islets/cell lines and human islets and hence, its low in terms of novelty. Further, based on authors' secondary analyses Il6 shows negative correlation with Pdx1, neurod1 etc. But upon silencing Il6, surprisingly these genes are down regulated. Insulin secretion by the human islets is not normalized to insulin content.

Author’s Response:

"Most of the work reported in the manuscript is already well known in rodent islets/cell lines and human islets and hence, it's low in terms of novelty”.

Previous studies exploring the role of IL6 have yielded inconsistent and sometimes contradictory results. For instance, Oliveira et al. (2023) demonstrated that treatment with IL-6 alone does not confer benefits to β-cells; instead, it leads to increased cell death markers and impaired activation of the unfolded protein response (UPR), partially attributable to endoplasmic reticulum (ER) stress, along with other mechanisms. Conversely, Paula et al. (2015) found that IL-6 signaling enhances the viability of pancreatic β-cells in response to physical exercise.

In our investigation, we adopted a comprehensive approach, combining bioinformatics analysis of publicly available data with in vitro experiments and clinical investigations to provide a better understanding of the role of IL6 in β-cell function. Our methodology encompassed various experimental techniques, including siRNA silencing, IL6 treatment, and assessments of glucose uptake, cell viability, apoptosis, and the expression of key β-cell genes. Our findings revealed a nuanced relationship between IL6 expression and specific key islet genes. Specifically, we observed negative correlations with PDX1, MAFB, and NEUROD1, while positive correlations were noted with SOX4, HES1, and FOXA1. This underscores the complexity of IL6's actions on β-cells, suggesting that these genes may encompass both causative and compensatory roles. Further elucidation of these pathways is warranted through additional studies.

Furthermore, we evaluated the gene expression patterns and investigated the functional impact of IL6 by examining glucose uptake. Our comprehensive approach sheds light on the multifaceted interplay between IL6 and β-cell function, providing valuable insights for further research in this field.

“Further, based on the authors' secondary analyses, Il6 shows a negative correlation with Pdx1, neurod1, etc. But upon silencing Il6, surprisingly, these genes are down-regulated."

The negative correlation between IL6 expression and Pdx1/Neurod1 seems unexpected, considering that IL6 is upregulated in diabetic islets and that seems to increase insulin secretion in INS-1 cells. In general, the results of transcriptomics analysis do not cope 100% with the q-PCR in studies. The co-expression relationship between different genes may be influenced by several factors such as medication, genetics, the local environment or metabolic heterogeneity (glucotoxicity, lipotoxicity, hyperinsulinemia and, insulin resistance etc.), among other factors1,2.

In this context, our data demonstrated that IL6 expression in human islets could be attributed to other factors such as glucotoxicity, lipid toxicity or steroid hormones, rather than a direct cause. “Interestingly, recent reports have revealed an increased expression of IL6 in response to insulin administration in 3T3-L1 adipocytes [35]. Although these findings have been reported in adipose tissue, it may explain that increased expression of IL6 mRNA is a sequence of hyperinsulinemia associated with insulin resistance. To shed more light on this issue, we investigated the impact of exposure to glucose, lipids, steroid hormones, and antidiabetic medications on IL6 expression in INS-1 cells. It was evident that short-term exposure to high glucose levels, PA or Dexamethasone for 24 hours increased IL6 expression, denoting the role of IL6 as a mediator of the effect of those factors on β-cells. In contrast, antidiabetic medications (insulin, metformin, and Rosiglitazone) showed no impact on IL6 expression. These data indicate that the upregulation of IL6 in diabetic islets may be attributed to glucotoxicity, lipid toxicity or steroid hormones rather than a direct cause. Our findings also revealed that IL6 expression was not changed under short-term insulin exposure; assuming that IL6 elevation may start during the early phases of insulin resistance (and hyperinsulinemia), longer exposure of insulin is needed to elevate IL6 expression or treatment with different medications in vitro has differential impact compared to in vivo administration.”

On the other hand, It's possible that when Il6 is silenced, cells activate compensatory mechanisms to maintain cellular homeostasis. These compensatory mechanisms might inadvertently down-regulate the expression of genes like Pdx1 and neurod1, which could be part of the same regulatory network or pathway. Another possible explanation is Feedback Loops, where one gene's expression influences others' expression. Silencing Il6 might disrupt these feedback loops, leading to unexpected changes in gene expression.

1- Ottosson-Laakso E, Krus U, Storm P, et al. Glucose-induced changes in gene expression in human pancreatic islets: causes or consequences of chronic hyperglycemia. Diabetes. 2017;66(12):3013-3028.

2- Niu Y, Lin Z, Wan A, et al. RNA N6-methyladenosine demethylase FTO promotes breast tumor progression through inhibiting BNIP3. Molecular cancer. 2019;18(1):1-16.

“Insulin secretion by the human islets is not normalized to insulin content.”

Results from human islets were normalized for protein content, not for insulin content. This has been stated in the figure and legend in the revised version.

Reviewer 2 Report

Comments and Suggestions for Authors

In this manuscript, the authors used rat insulinoma cell line (INS-1 cells) and human islets from healthy and T2D subject to investigate the impact of IL-6 on insulin secretion. They reported that IL-6 gene expression and is highly expressed in islet from diabetic and old donors when compared to young and healthy donors. They also reported that silencing il6 gene in INS-1 cells reduced insulin secretion and glucose uptake and was associated with Insulin, PDX1, 23 NEUROD1 and GLUT2. However, IL-6 treatment increased insulin secretion both in INS-1 cells and human islets

The manuscript is a well written and present some interesting data regarding the role of IL-6 on pancreatic islet insulin secretion in healthy and T2D which remain controversial.

 General comment:

 The authors present data where they show that IL-6 expression correlated negatively with PDX1, MAFB, 20 NEUROD1 and positively with SOX4, HES1, and FOXA1. However, using INS-1 cells line they showed that IL-6 siRNA downregulates Ins2, PDX1, NEUROD1 and Glut 2 whereas IL-6 treatment increased Ins1 but has no major effect on PDX, NUROD1 and Glut2. Therefore, these results do not support their conclusion. I suggest to the authors to revise their conclusions and provide more explanation of these discrepancies between human islets and INS-1 cell line. I am not whether presenting figure from (https://mae.crc.med.lu.se/IsletGeneView/) is permitted or not. (This has to be verified as well. 

Specific comments:

The abstract need to be revised according to the general comments above.

Line 159: 

The authors wrote: SYBR green gene expression analysis for several β-cell function genes.

No CYBR gene analysis was performed. This sentence should read “SYBR green qPCR analysis for several β-cell function genes..”

Line 170: Reference (Taneera, Mohammed, Mohammed et al., 2020) is either wrong or not cited in the reference section. Please check and correct.

Line 215, Figure 1E-1H: 

Please specify in mRNA or protein levels are analyzed.

 Line 2020, figure 1F, 

 What is the status of these donors diabetic or not diabetic. Please clarify.

Line 316: 

How the authors explain a decrease of Pdx1 and NeuroD1 that were found negatively correlated with IL6 in human islets..

 Moreover, reduction of mRNA expression of Snap25 does not correlate with it protein expression level. Please comments.

Line 335, Figure 5:

How the authors explain no changes in glucose uptake and increased insulin release following Il6 treatment. Moreover, Il6 reduced glut2 expression. Did the authors analyze cell surface and cytoplasmic glut2 which may explain their results.

Line 354, Figure 6:

Again, certain genes that are negatively correlate with il6 in human islets are not affected by Il6 treatment. One expects, at least a downregulation of some of those genes when il6 treatment is applied. Please comment and explain.

Line 368:

Please change IL6 tretment for IL6 treatment

Lines 483 -488 in discussion section:

Opposite results were found in this manuscript using INS-1 mouse cell line. This has to be discussed and explained in the discussion section.

Comments on the Quality of English Language

English is good. I have no concern.

Author Response

Reviewer #2: In this manuscript, the authors used rat insulinoma cell line (INS-1 cells) and human islets from healthy and T2D subject to investigate the impact of IL-6 on insulin secretion. They reported that IL-6 gene expression and is highly expressed in islet from diabetic and old donors when compared to young and healthy donors. They also reported that silencing il6 gene in INS-1 cells reduced insulin secretion and glucose uptake and was associated with Insulin, PDX1, 23 NEUROD1 and GLUT2. However, IL-6 treatment increased insulin secretion both in INS-1 cells and human islets. The manuscript is a well written and present some interesting data regarding the role of IL-6 on pancreatic islet insulin secretion in healthy and T2D which remain controversial.

General comment: 

The authors present data where they show that IL-6 expression correlated negatively with PDX1, MAFB, 20 NEUROD1 and positively with SOX4, HES1, and FOXA1. However, using INS-1 cells line they showed that IL-6 siRNA downregulates Ins2, PDX1, NEUROD1 and Glut 2 whereas IL-6 treatment increased Ins1 but has no major effect on PDX, NUROD1 and Glut2. Therefore, these results do not support their conclusion. I suggest to the authors to revise their conclusions and provide more explanation of these discrepancies between human islets and INS-1 cell line. I am not whether presenting figure from (https://mae.crc.med.lu.se/IsletGeneView/) is permitted or not. This has to be verified as well. 

Author’s Response: We thank the reviewer for the comment. Indeed, we added new paragraph about this issue in the discussion “The opposite findings observed in this study between INS-1 cells and human islets can predominantly be attributed to species variation. Such variations suggest that some degree of species-specific traits may exist or that there are disparities between freshly isolated normal cells and immortalized cell lines. Hence, selecting the appropriate experimental model is crucial when aiming to accurately characterize the functional aspects of gene/receptor and their impact on cell biology.”

The Geneview tool is free to access. The developers have added this statement on their website : “To cite data from this tool, please cite: "Islet Gene View - a tool to facilitate islet research" ,Asplund et.al, Life Science Alliance 2022 using the link below. https://doi.org/10.26508/lsa.202201376”. We have inserted the reference into the revised MS.

Specific comments:

  • The abstract need to be revised according to the general comments above.

Author’s Response:  Done. We revised the conclusion in the abstract as advised.

“This study provides a new understanding of the role of IL6 in β-cell function and the pathophysiology of T2D. Our data highlights differences in the response to IL6 between INS-1 cells and human islets, suggesting the presence of species-specific variations across different experimental models. Further research is warranted to unravel the precise mechanisms underlying the observed effects of IL-6 on insulin secretion.”

  • Line 159: The authors wrote: SYBR green gene expression analysis for several β-cell function genes. No CYBR gene analysis was performed. This sentence should read “SYBR green qPCR analysis for several β-cell function genes.”

Author’s Response: The sentence has been corrected as suggested by the reviewer

  • Line 170: Reference (Taneera, Mohammed, Mohammed et al., 2020) is either wrong or not cited in the reference section. Please check and correct.

Author’s Response: Reference (Taneera, Mohammed, Mohammed et al., 2020) has been added. “Taneera J, Mohammed I, Mohammed AK, Hachim M, Dhaiban S, Malek A, Dunér P, Elemam NM, Sulaiman N, Hamad M, Salehi A. Orphan G-protein coupled receptor 183 (GPR183) potentiates insulin secretion and prevents glucotoxicity-induced β-cell dysfunction. Molecular and cellular endocrinology. 2020 Jan 1;499:110592.”

  • Line 215, Figure 1E-1H:  Please specify in mRNA or protein levels are analyzed.

Author’s Response: in Line 215 and Figure 1E-1H, we added the word “mRNA”.

  • Line 2020, figure 1F, What is the status of these donors diabetic or not diabetic. Please clarify.

Author’s Response: The donors are mixed (diabetic/non-diabetic), we clarified in the revised manuscript.

  • Line 316: 

How the authors explain a decrease of Pdx1 and NeuroD1 that were found negatively correlated with IL6 in human islets. Moreover, reduction of mRNA expression of Snap25 does not correlate with it protein expression level. Please comments.

Author’s Response: The negative correlation between IL6 expression and Pdx1/Neurod1 seems unexpected, considering that IL6 is upregulated in diabetic islets and that seems to increase insulin secretion in INS-1 cells. The co-expression relationship between different genes may be influenced by several factors such as medication, genetics, the local environment or metabolic heterogeneity (glucotoxicity, lipotoxicity, hyperinsulinemia and, insulin resistance etc.), among other factors1,2. In this context, our data demonstrated that IL6 expression in human islets could be attributed to other factors, such as glucotoxicity, lipid toxicity or steroid hormones, rather than a direct cause. Additionally, the reduction in Pdx1 and NeuroD1 expression could result from downstream effects of IL6 signaling. IL6 may modulate the activity of transcription factors or signaling pathways that influence Pdx1 and NeuroD1 expression.

The possible explanation of the discrepancy between Snap25 mRNA and protein expression is the complexity of post-transcriptional and translational regulatory processes, for example mRNA stability, translational efficiency, protein degradation or regulatory mechanisms disturbing protein synthesis or turnover.

1- Ottosson-Laakso E, Krus U, Storm P, et al. Glucose-induced changes in gene expression in human pancreatic islets: causes or consequences of chronic hyperglycemia. Diabetes. 2017;66(12):3013-3028.

2- Niu Y, Lin Z, Wan A, et al. RNA N6-methyladenosine demethylase FTO promotes breast tumor progression through inhibiting BNIP3. Molecular cancer. 2019;18(1):1-16.

  • Line 335, Figure 5: How the authors explain no changes in glucose uptake and increased insulin release following Il6 treatment. Moreover, Il6 reduced glut2 expression. Did the authors analyze cell surface and cytoplasmic glut2 which may explain their results.

Author’s Response: We appreciate the reviewer's comment. We are afraid of misunderstanding this issue as our data do not show any significant reduction in Glut2 expression at mRNA or protein levels after IL-6 treatment (Fig. 5).

  • Line 354, Figure 6: Again, certain genes that are negatively correlate with il6 in human islets are not affected by Il6 treatment. One expects, at least a downregulation of some of those genes when il6 treatment is applied. Please comment and explain.

Author’s Response: Please see our response on comment # 6.

  • Line 368: Please change IL6 tretmentfor IL6 treatment

Author’s Response: The sentence has been changed as the reviewer suggested.

  • Lines 483 -488 in discussion section: Opposite results were found in this manuscript using INS-1 mouse cell line. This has to be discussed and explained in the discussion section.

Author’s Response: We thank the reviewer for providing this insightful comment. The opposite findings observed in this study can predominantly be attributed to species variation, namely the disparity between the INS-1 rat cell line and human islets. Such variations suggest that some degree of species-specific traits may exist or that there are disparities between freshly isolated normal cells and immortalized cells. Should the former scenario hold true, it suggests potential differences in the functional significance of various genes/receptors across distinct species.

We added a paragraph about this issue to accommodate the reviewer's comment in the discussion part. “The opposite findings observed in this study between INS-1 cells and human islets can predominantly be attributed to species variation. Such variations suggest that some degree of species-specificity traits may exist or that there are disparities between freshly isolated normal cells and immortalized cell line.”

Reviewer 3 Report

Comments and Suggestions for Authors

Several minor remarks.

Lines 21-23 "Silencing IL6 in INS-1 cells reduced  insulin secretion and glucose uptake without independent of any involment od apoptosis or oxidative stress" should be changed to "Silencing of IL6 in INS-1 cells should be changed to "Silencing IL6 in INS-1 cells reduced insulin secretion and glucose uptake independently of apoptosis or oxidative stress".

Lines 106-107 "RNA and protein were extpracted and subjected to mRNA or protein expression" should be changed to "RNA and protein were extracted and subjected to to mRNA or protein quantification". 

Author Response

Reviewer #3: Several minor remarks.

  • Lines 21-23 "Silencing IL6 in INS-1 cells reduced insulin secretion and glucose uptake without independent of any involment od apoptosis or oxidative stress" should be changed to "Silencing of IL6 in INS-1 cells should be changed to "Silencing IL6 in INS-1 cells reduced insulin secretion and glucose uptake independently of apoptosis or oxidative stress".

Author’s Response: The statement has been changed as the reviewer suggested.

  • Lines 106-107 "RNA and protein were extpracted and subjected to mRNA or protein expression" should be changed to "RNA and protein were extracted and subjected to to mRNA or protein quantification". 

Author’s Response: The sentence has been changed as the reviewer suggested.

Reviewer 4 Report

Comments and Suggestions for Authors

In this study, Taneera et al. analyze the effects of T2D on serum levels of IL6 in patients, and downstream effects in INS-1 cells. Understanding the contribution of altered chemokine levels in T2D might help to identify important markers that are associated with disease severity and risk of serious complications. Using publicly available RNAseq databases, the authors analyze mRNA levels of proposed islet markers in T2D patients. The authors then conduct qPCR measurements to identify mRNA levels of a gene panel in INS-1 cells, either after IL6 knockdown, or stimulated with IL6, and they conduct a range of cell assays. Finally, they analyse correlations from patient sera and correlate these with clinical parameters.

The main findings of the manuscript are: (1) enhanced islet IL6 levels in T2D diseased patients in silico, (2) shRNA knockdown of IL6 leads to altered islet gene expression, while (3) IL6 administration led to reversed effects of gene expressions. Moreover, (4) enhanced serum IL6 levels correlated with T2D in untreated patients, and (5) IL6 levels were unchanged in patients undergoing T2D therapy. The authors point out the specific functional relevance of IL6 as a persistent inflammatory driver in T2D.

Comment: The study is strongly connected to previous work of the authors in the last 5 years that is conducted completely repetetive. Following the same scheme, the authors now interrogate the influence of IL6, showing IL6 presence and that T2D leads to changes of specific islet gene profiles. Analyses and experiments conducted in the manuscript follow robust methods and are well controlled. The methods and results are described detailed. Although this paper follows exactly the same pattern as the ten publications of the previous 5 years, important labels and descriptions are missing. A further shortcoming of the manuscript is that there is a lack of argumentation and clear interpretation of results, following no hypothesis. Publication can only be considered after essential changes.  

Major comments:

1) Results: introducing and summarizing sentences should be amended to generate a sequence of scientific argumentation.

2) Figure legends: the inscriptions are not very informative, as they do not describe the key result of the experiments.

3) Results 3.1, Fig.1A, lane 202f:  describe the analyzed patient groups: how many were included? Indiate in Fig.1A, that these were healthy subjects. Could you undertake a comparison with T2 islets?

4) Results 3.1., Fig.1B-D, lane 210f: what do these results tell us? This is all data from healthy patients, it should be compared to diabetic ones.

5) Figure panels 1B-H should be enlarged, and also its inscriptions.

6) Results 3.2, Fig. 2, lane 259f: what do these gene and transcription factor expression changes tell us about beta-cell function or role of IL6? What is the conclusion from these gene expression data? Is the islet function enhanced or reduced? Please build a coherent hypothesis.

7) lane 263f: how are these stimulation experiments connected to the previous data? Maybe better show the other way round.

8) Results 3.3 lane 283: explain in an introductory sentence why you aimed to perform IL6 knockout. What is your hypothesis?

9) Results 3.3, lanes 292,294,296: wrong Figure panels in the text: 3E,3F,3G instead of 2E,2F,2G.

10) Results 3.3, Fig. 2G, lane 295: how was the insulin content determined, method is not described.

11) Results 3.4, lane 313: here the authors start with an arguing sentence, but they do not specify in which cells they aim to study the IL6 role.

12) Results 3.4, lane 321: what do the observed protein expression reductions mean? How does this add to a working hypothesis? Is islet function disturbed or unchanged?

13) Results Fig.6 legend: which cells were studied? Please amend. What is the overall interpretation of these results in relation to the IL6 knockdown?

14) Results 3.6, Fig.7A, lane371f: Please clarify, which islets are shown: are these two T2D derived islets? what does the difference between the red and black bar with high glucose stimulation indicate? Is the red one a healthy control islet?

15) Results 3.6, Fig.7C, lane 386: Fig.7C instead of 7B. lane 397: in the Fig.7 legend, (C) is missing.

16) Results 3.6, lane 389: amend “age and gender”, amend “Table 3, right column”, and “only fasting glucose etc…

17) lane 390: an overall concluding remark is missing. What does this all mean? What is your conclusion?

18) Discussion/Conclusion: Do the authors have an idea for the significance of IL6 in therapeutic terms?

Which perspective do you see for your results to be used in diagnosis and/or therapy?

Minor comments:

1) Figure inscriptions are generally very small and hardly readable. Please enlarge.

Comments on the Quality of English Language

English level: good, but can be improved

Author Response

Reviewer #4:

In this study, Taneera et al. analyze the effects of T2D on serum levels of IL6 in patients, and downstream effects in INS-1 cells. Understanding the contribution of altered chemokine levels in T2D might help to identify important markers that are associated with disease severity and risk of serious complications. Using publicly available RNAseq databases, the authors analyze mRNA levels of proposed islet markers in T2D patients. The authors then conduct qPCR measurements to identify mRNA levels of a gene panel in INS-1 cells, either after IL6 knockdown, or stimulated with IL6, and they conduct a range of cell assays. Finally, they analyse correlations from patient sera and correlate these with clinical parameters.

The main findings of the manuscript are: (1) enhanced islet IL6 levels in T2D diseased patients in silico, (2) shRNA knockdown of IL6 leads to altered islet gene expression, while (3) IL6 administration led to reversed effects of gene expressions. Moreover, (4) enhanced serum IL6 levels correlated with T2D in untreated patients, and (5) IL6 levels were unchanged in patients undergoing T2D therapy. The authors point out the specific functional relevance of IL6 as a persistent inflammatory driver in T2D.

Comment: The study is strongly connected to previous work of the authors in the last 5 years that is conducted completely repetetive. Following the same scheme, the authors now interrogate the influence of IL6, showing IL6 presence and that T2D leads to changes of specific islet gene profiles. Analyses and experiments conducted in the manuscript follow robust methods and are well controlled. The methods and results are described detailed. Although this paper follows exactly the same pattern as the ten publications of the previous 5 years, important labels and descriptions are missing. A further shortcoming of the manuscript is that there is a lack of argumentation and clear interpretation of results, following no hypothesis. Publication can only be considered after essential changes.  

Major comments:

  • Results: introducing and summarizing sentences should be amended to generate a sequence of scientific argumentation.

Author’s Response: We thank the reviewer for capturing our attention for this observation. As requested, we amended most of the introductory and summarizing sentences in the results section.

  • Figure legends: the inscriptions are not very informative, as they do not describe the key result of the experiments.

Author’s Response: Thank you so much for the comment. We improved the inscriptions as the reviewer advised. For example “ Figure 3. Silencing of IL6 expression causes β-cell dysfunction in INS-1 cells”

  • Results 3.1, Fig.1A, lane 202f:  describe the analyzed patient groups: how many were included? Indiate in Fig.1A, that these were healthy subjects. Could you undertake a comparison with T2 islets?

Author’s Response: Adscription are already mentioned in figure1 legend “Expression pattern of IL6 and IL6R in human pancreatic islets. (A) RNA-seq expression of IL6 and IL6R compared to other key pancreatic β-cell genes (KCNJ11, PDX1, GCK, GLUT1, MAFA, SNAP25, NEUROD1, INSR and VAMP2) in non-diabetic human pancreatic islets (n = 68)”.

However, as the reviewer advised, we have added some detailed descriptions in the result section 3.1. “Figure 1A, our findings reveal the presence of IL6 and IL6R transcripts in non-diabetic human pancreatic islets (n=68).”

For the last question, we have shown similar data in different publications, such as

  • Identification of novel genes for glucose metabolism based upon expression pattern in human islets and effect on insulin secretion and glycemia. Human Molecular Genetics, Volume 24, Issue 7, 1 April 2015, Pages 1945–1955, https://doi.org/10.1093/hmg/ddu610.
  • Global genomic and transcriptomic analysis of human pancreatic islets reveals novel genes influencing glucose metabolism.111 (38) 13924-13929. https://doi.org/10.1073/pnas.1402665111.
  • A Systems Genetics Approach Identifies Genes and Pathways for Type 2 Diabetes in Human Islets. Cell Metabolism 2012. https://www.cell.com/cell-metabolism/pdf/S1550-4131(12)00243-4.pdf.

However, as you can see in the attached figure, only KCNJ11, PDX1 and NEUROD1 expression were significantly reduced in diabetic islets compared to non-diabetic islets.

  • Results 3.1., Fig.1B-D, lane 210f: what do these results tell us? This is all data from healthy patients, it should be compared to diabetic ones.

Author’s Response: We agree with the reviewer that comparing data in Figure 1B- and D in healthy vs. diabetic would be great information. However, these data were obtained from the GeneVeiw tool, and sadly, they only have data from nondiabetic tissues.

The results of our MS tell us that IL6 is highly expressed in human islets and fat cells compared to the liver and skeletal muscles (metabolic tissues). In contrast, IL6R was highly expressed in muscle and liver. This finding highlights that IL6 and IL6R are expressed differentially across different tissues. We explained these results in detail in the discussion part (see below)

“The observed expression patterns of IL6 and IL6R in different metabolic tissues likely reflect a distinct function and responses to various environmental and physiological cues, such as immune responses, inflammation, and metabolic processes. For example, IL-6 may be expressed in islets as part of the immune response or as a signaling molecule regulating insulin secretion or glucose metabolism. Fat cells secrete various signaling molecules, including cytokines like IL-6, as a part of their involvement in inflammation, metabolism, and energy homeostasis. The high expression of IL-6R in muscle may be related to its role in muscle function and repair. In contrast, it may be associated with regulating the response to inflammation or injury in the liver and muscles.”

Figure 1D: It is widely believed that IL6 exerts its effects by binding to its receptor, IL6R, and activating a complex signaling pathway that modulates the expression and activity of various transcription factors. Therefore, we aim to address this relation in human pancreatic islets. Our data showed no correlation between IL6 and IL6R, suggesting that IL-6 doesn’t play an autocrine function.

  • Figure panels 1B-H should be enlarged, and also its inscriptions.

Author’s Response: We apologize for this issue. We enlarged the figure panels 1B-H and its inscriptions.

  • Results 3.2, Fig. 2, lane 259f: what do these gene and transcription factor expression changes tell us about beta-cell function or role of IL6? What is the conclusion from these gene expression data? Is the islet function enhanced or reduced? Please build a coherent hypothesis.

Author’s Response: Our analysis emphasizes the complex and varied nature of IL6's interactions with genes that play pivotal roles in pancreatic β-cell function, encompassing transcription regulation, exocytosis, and calcium transport. Based on our data, we can say that the physiological range of IL6 enhanced the islet's function; similarly, IL6 knockdown reduced the islet function.

  • lane 263f: how are these stimulation experiments connected to the previous data? Maybe better show the other way round.

Author’s Response: Regarding the previously published data on factors influencing gene expression, such as glucotoxicity, lipotoxicity, steroid hormones, and medications, we agree that certain factors may play a role in IL6 expression. To investigate the effect of these factors on Il6 expression at the mRNA level in INS-1 cells, we conducted simulation experiments using glucose, palmitic acid, dexamethasone, insulin, metformin, and rosiglitazone. Our results indicate that a 24-hour exposure to glucose, dexamethasone, and palmitic acid significantly enhanced Il6 expression in INS-1 cells. These findings support the notion that these factors can influence IL6 expression, potentially contributing to the pathophysiology of diabetes.

We apologize for any confusion caused by the reviewer's second point. We appreciate their suggestion to explore alternative stimulation experiments. However, given our study's specific aims and design, it is not feasible to perform the stimulation experiments in the reverse manner as suggested. We believe that the experiments and results provided valuable insights into the regulation of Il6 expression in INS-1 cells.

  • Results 3.3 lane 283: explain in an introductory sentence why you aimed to perform IL6 knockout. What is your hypothesis?

Author’s Response: As requested by the reviewer, an introductory sentence was inserted in results 3.3. in the revised version “To study the precise role of IL6 in the pancreatic beta cells, we eliminate the expression of IL6 in INS-1 cells to elucidate its role on β-functions, insulin secretion, glucose uptake and gene expression regulation”

  • Results 3.3, lanes 292,294,296: wrong Figure panels in the text: 3E,3F,3G instead of 2E,2F,2G.

Author’s Response: We apologize for this error. In the revised manuscript, we corrected the figure panels in the text.

  • Results 3.3, Fig. 2G, lane 295: how was the insulin content determined, method is not described.

Author’s Response: A detailed method for insulin content has been added into the methods section based on the reviewer's comment. “To measure the insulin content, the total protein in the cells was extracted and diluted at 1:250. The insulin content was then assessed using an ELISA assay. Finally, results were normalized to the total protein amount in the cells.”

  • Results 3.4, lane 313: here the authors start with an arguing sentence, but they do not specify in which cells they aim to study the IL6 role.

Author’s Response: We apologize for this. We mentioned the name of cells (INS-1) in the 3.4 result section.

  • Results 3.4, lane 321: what do the observed protein expression reductions mean? How does this add to a working hypothesis? Is islet function disturbed or unchanged?

Author’s Response: Silencing of Il6 resulted in reduced protein expression of key genes in pancreatic β-cells such as INSULIN, PDX1, MAFA and GLUT2. Importantly, this downregulation protein expression was associated with impaired insulin secretion and reduced glucose uptake without affecting apoptosis levels or ROS production.

The impairment of insulin secretion was associated with the downregulation of key β-cell genes, including Insulin, Pdx1, NeuroD1 and Glut2, which further supported the crucial role of Il6 in maintaining normal β-cell function. Pdx1 and NeuroD1 are key transcription factors for β cell maturation and responsiveness to glucose through regulating the insulin gene expression. Glut2 gene is the main glucose transport in rodent β-cells and is a vital component of the glucose uptake machinery for insulin release. The finding indicates the importance of physiological Il6 levels for optimal β-cell function as reduced expression of IL6 may disturb islets function.

  • Results Fig.6 legend: which cells were studied? Please amend. What is the overall interpretation of these results in relation to the IL6 knockdown?

Author’s Response: We used INS-1 cells in figure 6. Our analysis of the effects of IL6 treatment on INS-1 cells highlights a significant upregulation in Pro/INSULIN, which provides insight into the mechanism behind the observed increase in insulin secretion post-IL6 treatment. Additionally, this experiment serves as validation for IL6 knockdown. In IL6-silenced cells, we noted a decrease in insulin secretion, correlating with the downregulation of PRO/INSULIN.

  • Results 3.6, Fig.7A, lane371f: Please clarify, which islets are shown: are these two T2D derived islets? what does the difference between the red and black bar with high glucose stimulation indicate? Is the red one a healthy control islet?

Author’s Response: We apologize for any confusion. The islets depicted in Figure 7A are isolated from two distinct non-diabetic/healthy donors, obtained from PRODO Lab in the USA. The red bar represents control human islets (without IL6 treatment), stimulated with either 2.6 mM glucose (representing basal insulin levels) or 16.7 mM glucose (representing stimulated insulin levels). The black bar signifies human islets treated with IL6 and subsequently stimulated with either 2.8 or 16.7 glucose concentrations. We clarify this in the revised version of the figure legend.

  • Results 3.6, Fig.7C, lane 386: Fig.7C instead of 7B. lane 397: in the Fig.7 legend, (C) is missing.

Author’s Response: Thanks for the comment. We added the missing part in the revised manuscript.

  • Results 3.6, lane 389: amend “age and gender”, amend “Table 3, right column”, and “only fasting glucose etc.

Author’s Response: Corrected.

  • lane 390: an overall concluding remark is missing. What does this all mean? What is your conclusion?

Author’s Response:  We appreciate the reviewer's comments. We revised this issue as the reviewer advised.

Next, we correlated IL6 levels with clinical variables in T2D-subjects. Correlation tests were performed in different models to assess whether these clinical variables have independent associations with IL6 levels. In model 1 (unadjusted analysis), IL6 levels were associated (p < 0.05) positively with age, WC, fasting glucose, HbA1c, and triglyceride levels (Table 3). Following adjustment for age and gender in model 2 analysis, the associations between IL6 levels and fasting glucose, HbA1c, and triglycerides remained positive, indicating that increased levels of IL6 was associated with the higher blood sugar and lipids profile. These findings suggest that IL6 may play a role in the pathophysiology of T2D, possibly through its associations with metabolic parameters.”

  • Discussion/Conclusion: Do the authors have an idea for the significance of IL6 in therapeutic terms? Which perspective do you see for your results to be used in diagnosis and/or therapy?

Author’s Response: We thank the reviewer for the fruitful comment. In the revised version, we added a new paragraph about IL6 therapeutic perspective.

“The IL6 antagonists are indicated in the treatment of rheumatological disorders and in patients with cytokine release syndrome, occurring as an adverse effect of chimeric antigen receptor T cell therapy [50]. Both tocilizumab and sarilumab are recombinant humanized anti–IL6 receptor monoclonal antibodies. In the context of diabetes mellitus, as mentioned earlier, IL-6 has been implicated in developing insulin resistance, a hallmark of type 2 diabetes. Furthermore, the intricate interplay between chronic low-grade inflammation and insulin resistance, as well as obesity, underscores the significance of IL-6 in the pathophysiology and disease progression. Consequently, targeting IL-6 or its downstream signaling cascades emerges as a promising therapeutic strategy for effectively managing diabetes mellitus, with particular emphasis on T2D.”

Minor comments:

  • Figure inscriptions are generally very small and hardly readable. Please enlarge

Author’s Response: Thanks for the comment. As requested, we enlarged the figure inscriptions in the revised manuscript.

Round 2

Reviewer 1 Report

Comments and Suggestions for Authors

All my concerns have been adequately addressed.

Author Response

Thanks 

Reviewer 2 Report

Comments and Suggestions for Authors

I am satisfied with the authors responses.

 I don't have other comments

Author Response

Thanks

Reviewer 4 Report

Comments and Suggestions for Authors

Review report to the Authors - R2:

The revised manuscript by Taneera et al. addresses several points in detailed manner. The authors have improved the methods, text and descriptions of passages, and they have included many important informations so that it is now clearer to read. Nonetheless, several important remarks have not been considered by the authors and therefore, I still cannot recommend publication of this article.

Comments and Suggestions:

1) Results 3.1, Fig.1A-D: still, the authors do not clearly describe and show that this first part of the figure only contains expression data from healthy subjects and should be indicated above e.g. Fig.1A.

2) Fig. legend 1E, essentially, the number n=? of islets in each group is missing.

3) Legend Fig.2D please mention INS-1 cells in the legend and in the figure panel 2D or y-axis.

4) Results 3.4, lane 340f: data described are incorrect: in Fig.4A, there is no Insr mRNA expression result shown, while in Fig.4B there is no GCK western blot result shown. Please correct to what was really done.

5) Still, the authors do not comment in this section on what the observed mRNA/protein expression reductions (INS, PDX1, NEUROD1, GLUT2) or non-changes (MAFA, INSRb, SNAP25,VAMP2) mean. The authors are testing these genes since at least 2019 (11 publications in PubMed), so I guess the have an opinion on the meaning of these changes, not just measure them.

6) in the Fig.7 legend, still (C) is missing.

Comments on the Quality of English Language

English language quality is fine.

Author Response

First, we would like to thank the reviewer for the valuable comments and feedback.

REVIEWERS' COMMENTS:

Comments and Suggestions:

1- Results 3.1, Fig.1A-D: still, the authors do not clearly describe and show that this first part of the figure only contains expression data from healthy subjects and should be indicated above e.g. Fig.1A.

  • Response: Corrected as advised.

 2- legend 1E, essentially, the number n=? of islets in each group is missing.

  • Response: Corrected as advised.

3- Legend Fig.2D please mention INS-1 cells in the legend and in the figure panel 2D or y-axis.

  • Response: Corrected as advised.

4- Results 3.4, lane 340f: data described are incorrect: in Fig.4A, there is no Insr mRNA expression result shown, while in Fig.4B there is no GCK western blot result shown. Please correct to what was really done.

  • Response: We apologize for this mistake. We revised the text as advised.

5- Still, the authors do not comment in this section on what the observed mRNA/protein expression reductions (INS, PDX1, NEUROD1, GLUT2) or non-changes (MAFA, INSRb, SNAP25,VAMP2) mean. The authors are testing these genes since at least 2019 (11 publications in PubMed), so I guess the have an opinion on the meaning of these changes, not just measure them.

  • Response: We thank the reviewer for this important point.

Measuring the expression of genes such as INS, PDX1, NEUROD1, GLUT2, MAFA, INSRb, SNAP25, and VAMP2 in pancreatic β-cells offers critical insights into the function of these pivotal cells in glucose homeostasis.

INS (Insulin) is the primary hormone synthesized by β-cells. PDX1 plays a vital role in pancreas development, specifically in specifying pancreatic endoderm and guiding subsequent differentiation into various pancreatic cell types, including beta cells. PDX1 also regulates genes involved in insulin biosynthesis and secretion, ensuring proper beta cell function.

NEUROD1, another essential transcription factor, is integral in the differentiation of pancreatic endocrine cells, including beta cells. It not only guides beta cell development but also maintains its identity and function, directly enhancing insulin gene expression.

MAFA contributes significantly to beta cell maturation, ensuring they acquire essential characteristics for optimal function. This transcription factor regulates insulin gene expression and fine-tunes glucose-stimulated insulin secretion, a critical aspect of beta cell physiology.

SNAP25 and VAMP2, known as synaptobrevin, are vital proteins involved in synaptic vesicle fusion during insulin secretion. These proteins facilitate the exocytosis of insulin-containing vesicles, allowing for the precise regulation of insulin release in response to glucose levels.

GLUT2 and GCK are indispensable for glucose metabolism regulation within beta cells. GLUT2 facilitates glucose uptake, enabling beta cells to sense even subtle changes in blood glucose levels. GCK acts as a glucose sensor, initiating insulin secretion upon glucose phosphorylation, ultimately leading to ATP-dependent insulin release.

INSRβ, the β cell-specific isoform of the insulin receptor, plays a central role in mediating insulin effects within β-cells. Activation of INSRβ triggers signaling cascades that regulate glucose uptake, glycogen synthesis, and cell growth. Additionally, INSRβ signaling is crucial for maintaining beta cell function and survival, protecting against apoptosis induced by various stressors.

Thus, understanding the roles of these genes/proteins in pancreatic β-cells is pivotal for unraveling the mechanisms underlying glucose homeostasis. Dysregulation of these factors can lead to β-cell dysfunction and contribute to diabetes pathophysiology, highlighting their significance in metabolic health.

In the revised MS, we added an introductory paragraph in section results 3.4: “Measuring gene expression in pancreatic β-cells provides insights into glucose homeostasis. For example, Ins, Pdx1, Neurd1, Mafa, Glut2, Gck, Insrβ, Snap25, and Vamp2 are key genes involved in regulating insulin production, secretion, and cell function. Dysregulation of these genes can lead to diabetes, emphasizing their importance in β-cell biology. Therefore, the impact of Il6 silencing on the expression of key genes crucial for β-cell function was investigated at transcriptional and translational levels in INS-1 cells.”

6- in the Fig.7 legend, still (C) is missing.

  • Response: We apologize for this mistake. We revised the text as advised.